# Dark matter and flavor anomalies with vector-like fermions and scalar leptoquark

Shivaramakrishna Singirala[1][*], Suchismita Sahoo[2] and Rukmani Mohanta[3]

**1** School of Physics, University of Hyderabad, Hyderabad 500046, India
**2** Department of Physics, Central University of Karnataka, Kalaburagi-585367, India
**3** School of Physics, University of Hyderabad, Hyderabad 500046, India
* krishnas542@gmail.com

November 23, 2021

## Abstract

We investigate vector-like fermionic dark matter and flavor anomalies in a simple extension of standard model, with doublet vector-like fermions of quark and lepton type and also a $S_1(\bar{3}, 1, 1/3)$ scalar leptoquark. An additional vector-like lepton singlet is included, whose admixture with vector-like lepton doublet plays the role of dark matter and is examined in relic density and direct detection perspective. We utilize the bounds from electroweak precision observables and also constrain the new couplings from the branching ratio and angular observables associated with $b \to sll(\nu_l \bar{\nu}_l)$, $b \to s\gamma$ decays.

## Introduction

The well proposed fundamental theory of elementary particles, Standard Model (SM), failed to explain the matter-antimatter asymmetry, existence of dark matter (DM) and the observation of tiny neutrino mass, which provide a clear indication of the presence of new physics (NP) beyond it.

In the last few years, a collection of interesting deviations from the SM has been manifested in various angular observables associated with flavor changing neutral currect (FCNC) $b \rightarrow sl^+l^-$ [1]. Most relevant anomalies include the lepton flavor universality violating (LFUV) parameters such as $R_K$ (2.5$\sigma$ deviation) [2], $R_{K^*}$ (2.2$\sigma$−2.4$\sigma$ deviation) [1]. The precise analysis of these deviations are needed in both the SM and beyond the SM scenarios in order to probe the structure of NP. On the other hand, linear behavior of galaxy rotation curves, bullet cluster etc strongly motivate the existence of DM. The primary motive of this paper is to make a combined study of DM and flavor anamalies on a single platform. For this, we extend SM with vector-like fermions and a $(\bar{3}, 1, 1/3)$ scalar leptoquark. We utilize the well-known singlet-doublet vector-like mixture of leptonic DM [3]. The vector-like fermions along with leptoquark provide new box diagrams for $b \rightarrow s$ transition, leading to a NP contribution in flavor sector.

The paper is arranged as follows. First we explain the model particle content, relevant interaction Lagrangian. Then we present relic density and DM-nucleon scattering of neutral vector-like DM in vaious portals. Followed by computation of electroweak precision parameters. Later, the constraints from the quark sector through $b \rightarrow sll(\nu_l \bar{\nu}_l)$ and $b \rightarrow s\gamma$ processes on model parameters is discussed. Finally, the conclusive remarks are provided.

## Model with vector-like fermions and leptoquark

We add SM with vector-like fermions i.e., quark doublet $\psi_q(\mathbf{3}, \mathbf{2}, 1/6) \equiv (\psi_u, \psi_d)^T$, lepton doublet $\psi_\ell(\mathbf{1}, \mathbf{2}, -1/2) \equiv (\psi_\nu, \psi_l)^T$ and a lepton singlet $\chi_\ell(\mathbf{1}, \mathbf{1}, 0)$. The scalar sector is introduced with a $S_1(\bar{3}, 1, 1/3)$ scalar leptoquark (SLQ). All the new fields are assigned with odd $Z_2$ charge. The new Lagrangian terms include

$$\mathcal{L} = -y_\ell \overline{Q_L^C} S_1 \epsilon^{ab} \psi_{\ell L} - y'_\ell \overline{d_R^C} S_1 \chi_{\ell R} - y_q \overline{\psi_{qL}^C} S_1 \epsilon^{ab} \ell_L - y'_q \overline{Q_L^C} S_1^* \epsilon^{ab} \psi_{qL} - y_D \overline{\psi_\ell} \tilde{H} \chi_\ell + \text{h.c.}$$

$$- M_q \overline{\psi_q} \psi_q - M_\psi \overline{\psi_\ell} \psi_\ell - M_\chi \overline{\chi_\ell} \chi_\ell + \overline{\psi_\ell} \gamma^\mu \left( i\partial_\mu - \frac{g}{2} \tau^a \cdot \mathbf{W}_\mu^a + \frac{g'}{2} B_\mu \right) \psi_\ell + \overline{\chi_\ell} \gamma^\mu \left( i\partial_\mu \right) \chi_\ell$$

$$+ \overline{\psi_q} \gamma^\mu \left( i\partial_\mu - \frac{g}{2} \tau^a \cdot \mathbf{W}_\mu^a - \frac{g'}{6} B_\mu \right) \psi_q + \left| \left( i\partial_\mu - \frac{g'}{3} B_\mu \right) S_1 \right|^2. \tag{1}$$

The term with coupling $y_D$ induces mixing between neutral vector-like leptons $\psi_\nu$ and $\chi_\ell$ and the lightest mass eigenstate ($N_2$) is a probable dark matter in the present model. The mass relations take the form

$$M_{\psi_\ell} = M_{N_1} \cos^2 \alpha + M_{N_2} \sin^2 \alpha, \quad M_{\chi_\ell} = M_{N_1} \sin^2 \alpha + M_{N_2} \cos^2 \alpha,$$
$$M_D = \Delta M \sin \alpha \cos \alpha, \tag{2}$$

with $\Delta M = (M_{N_1} - M_{N_2})$ represents the mass splitting between neutral mass eigenstates, plays a key role in relic density.

## Dark matter phenomenology

**Relic density :**
To compute the freeze-out abundance of vector-like neutral lepton, we use LanHEP [4] for model implementation and micrOMEGAs [5]. Annihilation channels mostly mediated by vector bosons. Other relevant parameters include mass splitting $\Delta M$, $y_\ell$ and $y'_\ell$. The mass splitting control the co-annihilation contribution and the Yukawa determines the impact of LQ portal channels, illustrated in the lower left panel of Fig. 1.

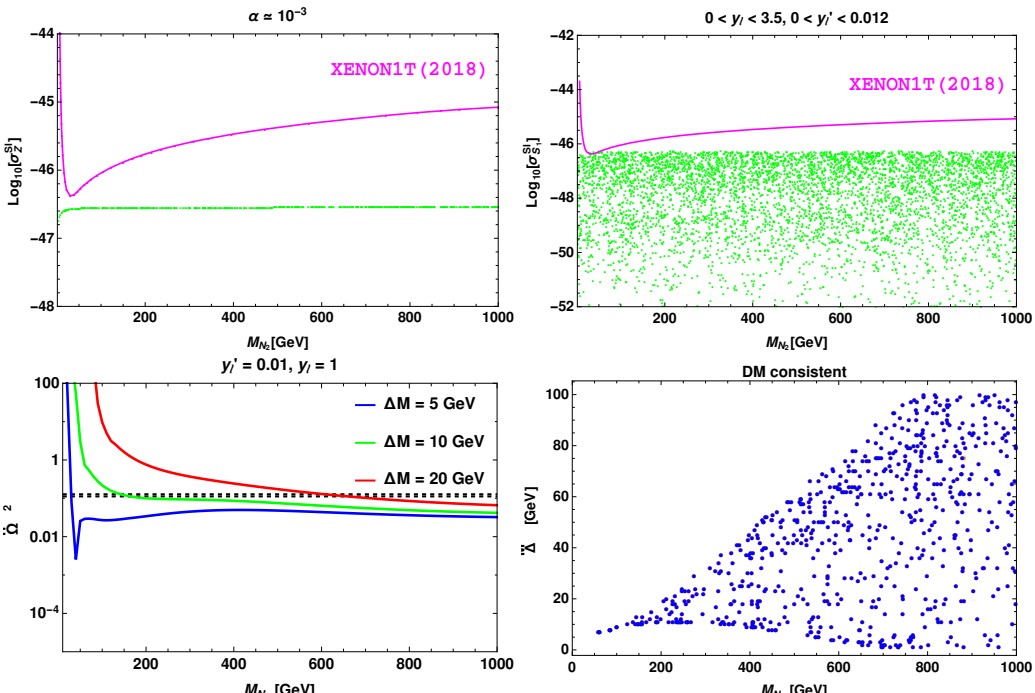

Figure 1: Upper panel projects SI WIMP-nucleon cross section in various portals, with magenta line correspond to XENON1T [6]. Lower left panel displays the behavior of relic density with horizontal dashed line representing Planck [7]. Lower right panel depicts the DM consistent parameter space in the plane of $\Delta M - M_{N_2}$.

**Direct detection :**
DM can scatter off nucleus to provide an imprint via $Z$ boson, LQ and Higgs. Effective Lagrangian of $Z$ and LQ portals contain vectorial form, which provide spin-independent cross section. XENON1T bound turn out to provide upper limit on the singlet-doublet mixing parameter and relevant Yukawa ($y'_l$) respectively, displayed in the upper panel of Fig. 1.

The DM consistent parameter space in view of above is displayed in the lower right panel of Fig. 1. One can see that larger mass splitting favor large DM mass spectrum.

## Electroweak precision parameters

The vector-like fermions in the present the model can alter the vacuum polarization of SM gauge bosons. Their gauge interactions can be studied through relevant parameters $\hat{S}$ and $\hat{T}$ [8]

$$\hat{S} = \frac{g}{g'} \times \Pi'_{W_3 B}(0), \quad \hat{T} = \frac{1}{M_W^2}\left(\Pi_{W_3 W_3}(0) - \Pi_{W^+ W^-}(0)\right), \tag{3}$$

and the relevant interaction lagrangian for vector-like leptons takes the form

$$\mathcal{L}_\ell \supset \left(\frac{g}{2}W_\mu^3 - \frac{g'}{2}B_\mu\right)\left(\overline{\psi_\nu}\gamma^\mu\psi_\nu\right) + \frac{g}{\sqrt{2}}W_\mu^+\left(\overline{\psi_\nu}\gamma^\mu\psi_l\right) + \frac{g}{\sqrt{2}}W_\mu^-\left(\overline{\psi_l}\gamma^\mu\psi_\nu\right). \quad (4)$$

No new constraint is levied on the model parameter space from the above parameters.

## Constraints from quark sector

The rare FCNC $b \to s$ transitions can proceed by the generic box diagrams in the presence of an additional scalar LQ and the vector-like fermion doublets, shown in Fig. 2.

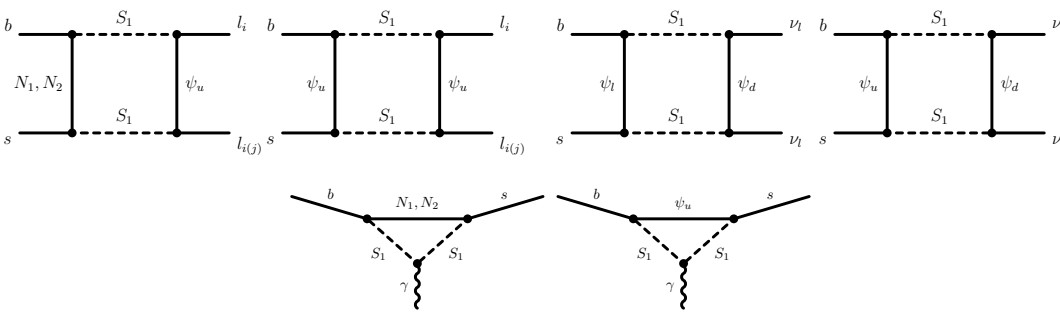

Figure 2: One-loop diagrams of $b \to s$ transition with scalar leptoquark and vector-like fermions in the loop.

### $b \to sll$

The effective Hamiltonian describing the $b \to sl^+l^-$ quark level transition is given by [9]

$$\mathcal{H}_{\text{eff}} = -\frac{4G_F}{\sqrt{2}}\lambda_t\left[\sum_{i=1}^{6}C_i(\mu)\mathcal{O}_i + \sum_{i=7,9,10}\left(C_i(\mu)\mathcal{O}_i + C_i'(\mu)\mathcal{O}_i'\right)\right], \quad (5)$$

where $C_i$'s are the Wilson coefficients and $\mathcal{O}_i$'s are their corresponding operators, given as

$$O_7^{(\prime)} = \frac{e}{16\pi^2}\left(\bar{s}\sigma_{\mu\nu}\left(m_s P_{L(R)} + m_b P_{R(L)}\right)b\right)F^{\mu\nu},$$
$$O_9^{(\prime)} = \frac{\alpha_{\text{em}}}{4\pi}(\bar{s}\gamma^\mu P_{L(R)}b)(\bar{l}\gamma_\mu l), \qquad O_{10}^{(\prime)} = \frac{\alpha_{\text{em}}}{4\pi}(\bar{s}\gamma^\mu P_{L(R)}b)(\bar{l}\gamma_\mu\gamma_5 l), \quad (6)$$

with $\alpha_{\text{em}}$ as the fine-structure constant and $P_{L,R}$ are the chiral operators. In the SM, the primed Wilson coefficients ($C_i'$) are zero, but they can have non-vanishing values in the NP models.

### $b \to s\nu_l\bar{\nu}_l$

The effective Hamiltonian of lepton flavor conserving $b \to s\nu_i\bar{\nu}_i$ process is given by [10]

$$\mathcal{H}_{\text{eff}}^{\nu\nu} = -\frac{4G_F}{\sqrt{2}}\lambda_t\, C_L^{\text{SM}}\mathcal{O}_L, \quad (7)$$

where

$$\mathcal{O}_L = \frac{\alpha_{\text{em}}}{4\pi}[\bar{s}\gamma^\mu P_L b][\bar{\nu}_i\gamma_\mu\left(1-\gamma^5\right)\nu_i], \quad (8)$$

is the six dimensional operator, $C_L^{\text{SM}} \approx -X(x_t)/\sin^2\theta_W$ is the SM Wilson coefficient calculated using the loop function $X(x_t)$ and $\theta_W$ is the weak mixing angle. Here $C_L^{ij}$ is zero in the SM.

$b \to s\gamma$

The effective Hamiltonian of $b \to s\gamma$ decay modes is given by

$$\mathcal{H}_{\text{eff}}^{\gamma} = -\frac{4G_F}{\sqrt{2}} V_{tb} V_{ts}^*(C_7^{\gamma\text{SM}} + C_7^{\gamma\text{NP}})\mathcal{O}_7 \,, \tag{9}$$

where $C_7^{\gamma\text{NP}} = -\dfrac{\sqrt{2}}{24 G_F V_{tb} V_{ts}^* M_{S_1}^2} \times \Big[|y_\ell \cos\alpha - y_\ell' \sin\alpha|^2 \tilde{F}_7(x_{N_1}) + |y_\ell \sin\alpha + y_\ell' \cos\alpha|^2 \tilde{F}_7(x_{N_2})$

$$+ |y_q'|^2 \big(\tilde{F}_7(x_u) + 2F_7(x_u)\big) \Big] \,, \tag{10}$$

$$\text{with } F_7(x) = \frac{x^3 - 6x^2 + 6x\log x + 3x + 2}{12(x-1)^4}, \quad \tilde{F}_7(x) = x^{-1}F_7(x^{-1}). \tag{11}$$

By using the recent measurements on the branching ratios of the processes such as $B_s \to ll$, $B \to K^{(*)}ll(\nu_l \bar{\nu}_l)$ [11], $\bar{B} \to X_s\gamma$ [12], and the lepton non-universality $R_{K^{(*)}}$ [1,2] parameters, we further constrained the new parameters like leptoquark couplings, vector-like fermions masses and couplings. The constrained region is displayed in Fig. 3.

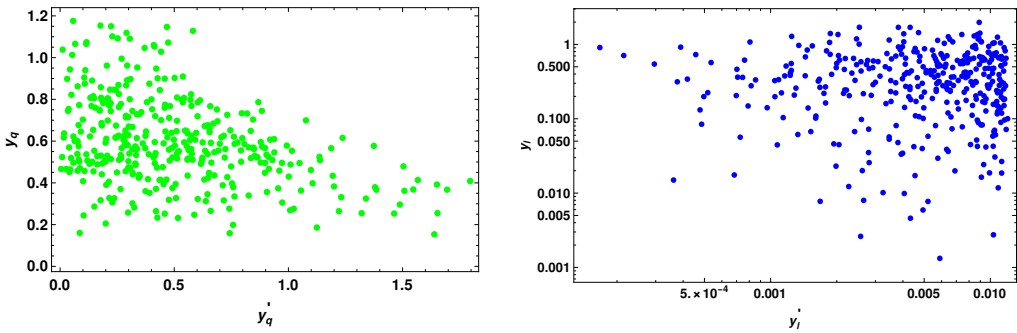

Figure 3: Constraints on $y_q' - y_q$ (left panel) and $y_\ell' - y_\ell$ (right panel) parameters obtained from the branching ratios of $B \to K^{(*)}l^+l^-(\nu_l \bar{\nu}_l)$, $B_s \to ll$ ($l = \mu, \tau$), $\bar{B} \to X_s\gamma$ processes, $R_{K^{(*)}}$ ratios.

## Conclusion

The model is motivated to shed light on dark matter and also the existing anomalies in flavor sector, associated with $B$-meson. So, we extend standard model with vector-like fermions of quark and lepton type and also with a $(\bar{3}, 1, 1/3)$ scalar leptoquark. An admixture of neutral vector-like lepton constitutes the relic density of the Universe through annihilation and co-annihilation channels mediated via scalar and gauge bosons. Apart from, the spin independent WIMP-nucleon cross section via $Z$-portal dictates the amount of mixing in neural vector-like leptons, and leptoquark-portal severely constrains the relevant Yukawa. In the presence of new vector-like fermions and scalar leptoquark, the rare $b \to s$ transitions occur through one loop box diagrams. By using the recent measurements on the branching ratios of $b \to sll(\nu_l \bar{\nu}_l)$ and $b \to s\gamma$ processes such as $B_s \to ll$, $B \to K^{(*)}ll(\nu_l \bar{\nu}_l)$, $\bar{B} \to X_s\gamma$, and the lepton non-universality $R_{K^{(*)}}$ parameters, we further constrained the new parameters like leptoquark couplings, vector-like fermions masses and couplings.

## Acknowledgements

S. Singirala and RM would like to thank University of Hyderabad for financial support through the IoE project grant IoE/RC1/RC1-20-012. RM acknowledges the support from SERB, Govt. of India through grant No, EMR/2017/001448.

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
