# Peer review of "Dark matter and flavor anomalies with vector-like fermions and scalar leptoquark"

_SciPost Physics Proceedings_

## Round 1 · Referee Report · Peisi Huang (Referee 1) · 2022-1-15

Report
In this proceeding, the authors address dark matter and the B-anomalies by extending the Standard Model (SM) with vector-like fermions, and a scalar leptoquark. With new fields assigned with odd $Z_2$ charges, the lightest state is a dark matter candidate. The authors constrain the parameter space by requiring the relic density is consistent with current observation, and the direct detection rates are within the current limit. Then, the authors discuss the possible explanations to B-anomalies using the vector-like fermions and the scalar leptoquark.
I have few questions to the authors,
1) I would expect contributions to muon g-2 from this scenario. How does the contributions to muon g-2 look like in the parameter space that is consistent with everything else?
2) In Fig 1, lower left panel, the authors choose specific values of yl and ylprime. How does the plot change when they go away from those values?
3) The authors never mentioned their choice of parameters for electron-leptoquark-quark couplings. Since it would be essential to $R_K^{(*)}$, and constraints from $B\rightarrow K^{(*)} e e$, I would recommend the authors clarify that part.
4) Few notations are not specified. For example, in Eq. (2), I understand those are the matrix element of the mass matrix, instead of the mass eigenvalues. It will be good clarify that to avoid confusion.

---

## Editorial Decision

unknown